

# Improving boundary layer flow simulations over complex terrain by applying a forest parameterization in WRF

**Johannes Wagner**[1], **Norman Wildmann**[1], **and Thomas Gerz**[1]

[1]Deutsches Zentrum für Luft- und Raumfahrt, Institut für Physik der Atmosphäre, Oberpfaffenhofen, Germany

**Correspondence:** Johannes Wagner (johannes.wagner@dlr.de)

**Abstract.** The impact of a forest parameterization on the simulation of boundary layer flows over complex terrain is investigated. Short- and long-term simulations are run for 12 hours and 1.5 months, respectively, with and without forest parameterization and the results are compared to lidar and meteorological tower observations. The test cases are based on the Perdigão 2017 campaign. Short-term simulations focus on low-level jet events over the double-ridge, while long-term simulations cover the whole intensive observation period of the campaign. Simulations without forest parameterization do not reproduce the interaction of the boundary layer flow with the double ridge satisfactorily. Surface winds are overestimated and flow separation and recirculation zones are not or only partly simulated. The additional drag of the forest parameterization considerably improves the agreement of simulated and observed wind speed and potential temperature by reducing the positive wind speed bias and increasing the correlation. The positive effect of the forest parameterization on the boundary layer flow is visible both in the short- and long-term simulations.





# 1   Introduction

Rising computational power allows to continuously increase grid resolutions of numerical weather models and enables to run real-case simulations with sub-kilometre meshes in large-eddy simulation (LES) mode. These simulations make it possible to study turbulent flows and their interaction with obstacles like mountains, forests, wind farms and single wind turbines in the atmospheric boundary layer. Real-case simulations are mostly performed with mesoscale models, which are designed to predict weather systems on the kilometre scale. This means that roughness elements near the surface are not explicitly resolved and are generally represented by areas of different roughness lengths $z_0$. This representation of surface friction becomes critical, when the horizontal and vertical model grid size is reduced to several 10 to 100 m, as obstacles like forests and buildings become partly resolved by the numerical model. The correct forecast of boundary layer flows and surface winds is of special interest for wind power predictions. As the number of wind turbines installed in complex and forested terrain continuously increases (e.g., Dellwik et al., 2014), the consideration of forest effects in numerical models is essential. Several modelling and measurement studies (e.g., Shaw and Schumann, 1992; Barr and Betts, 1997; Schröttle et al., 2016) analysed the effect of a forest parameterization on the boundary layer flow and showed that forest drag realistically describes boundary layer profiles of wind speed (e.g., inflection point profile), turbulent kinetic energy and turbulent momentum, heat and moisture fluxes.

In this study the forest parameterization of Shaw and Schumann (1992) is implemented in the WRF model and tested for flows over complex terrain. This parameterization has already been used by Wagner et al. (2019) and is described in more detail in this paper. The double ridge area of the Perdigão 2017 campaign, which is located in the Portuguese backcountry is used as test site due to the huge data set of meteorological observations (Fernando et al., 2019). Simulation results with and without forest parameterization will be compared to observations to quantify the effect of the forest friction. The paper is organized as follows: in section 2 a short overview of the Perdigão campaign and the used measurement data is given and the set-up of the numerical model and the implemented forest parameterization is described. Model results are compared to observations and an error quantification is provided in section 3. Conclusions and an outlook is finally given in section 4.





**Table 1.** Lidars and meteorological towers along TSE used to verify numerical simulations. For towers only data at the respective highest available altitude are used.

| Lidars | 100 m towers | 60 m towers | 30 m towers |
|--------|--------------|-------------|-------------|
| WS1 | T20 (tse04) | T22 (tse06) | T17 (tse01) |
| WS2 | T25 (tse09) | T27 (tse11) | T18 (tse02) |
| WS3 | T29 (tse13) | | T26 (tse10) |
| WS4 | | | |

## 2 Methodology

### 2.1 Campaign overview

The Perdigão campaign was a huge international field campaign to investigate the boundary layer flow in complex terrain and was part of the New European Wind Atlas (NEWA) project (Mann et al., 2017). It took place in the Portuguese backcountry and had an intensive observation period (IOP) from 1 May to 15 June 2017. The experimental site of Perdigão is characterized by mountainous terrain and a nearly parallel double ridge, which is oriented from northwest (NW) to southeast (SE) and has a valley depth and width of about 200 m and 1.4 km, respectively (see Fig. 1). The massive instrumentation with meteorological towers, lidars, microwave radiometers, radiosondes, wind profilers, radio acoustic sounding systems and microphones provided a unique data pool of meteorological observations in complex terrain (Fernando et al., 2019). On the SW ridge a single Enercon E-82 2 MW wind turbine (WT) with a hub height of 78 m and a rotor diameter of 82 m is located. This enabled to study the interaction of the boundary layer flow with the WT and the generated wakes under various meteorological conditions (e.g., Wildmann et al., 2018a, b; Barthelmie et al., 2018) The main wind directions at Perdigão are northeast (NE) and southwest (SW), which results in boundary layer flows that are nearly perpendicular to the double ridge. Frequently, low-level jet (LLJ) events were observed, which are mostly night-time phenomena. Wagner et al. (2019) could show that during the IOP LLJs were mainly thermally driven flows generated by the sourrounding mountainous area during synoptically calm conditions and that jets from NE occured more often than jets from SW. The complexity of the topography is increased by a patchy canopy layer of pine and eucalyptus forests (Menke et al., 2019a) with tree heights of 15 m to 20 m. Further details and a general overview of the field campaign can be found in Fernando et al. (2019).



## 2.2 Lidar-, tower- and radiosonde observations

In this study wind measurements of the DTU Doppler wind lidars WS1 to WS4 (Menke et al., 2018) and in-situ measurements of all 30 m, 60 m and 100 m meteorological towers along the southeast transect (TSE; equal to transect 2 in Fernando et al., 2019) are used to verify numerical simulations with and without forest parameterization. Radiosonde observations

(UCAR/NCAR - Earth Observing Laboratory, 2018) launched in the valley were used to visualize vertical wind conditions and to complement lidar observations. An overview of the used lidars and meteorological towers is given in Table 1 and their positions along TSE are shown in Fig. 1. A detailed view of the complete instrumentation of the Perdigão campaign is available in Fernando et al. (2019) and on the official Perdigão webpage[1]. The data repository of the field campaign is provided in the record re3data.org: Perdigao Field Experiment (2019). The four wind scanners WS1 to WS4 were performing range-height

indicator (RHI) scans across the double ridge parallel to a wind direction of 234.68°, which defines TSE. The scanning strategy was such that WS1 and WS3 were scanning towards SW using an azimuth angle of 234.68°, while WS2 and WS4 were scanning towards NE with an azimuth angle of 54.68°. Note that WS2 and WS3 are not shown in Fig. 1, as they are located close to WS1 and WS4, respectively. The combination of the four lidars enables to produce lidar composites of radial velocities perpendicular to the double ridge (Menke et al., 2019a). This direction is called "cross-valley" and the direction parallel to the

double-ridge "along-valley" in this study. Both directions are normal to each other and are indicated in Fig. 1.

In-situ measurements of horizontal wind speed and potential temperature at the meteorological towers listed in Table 1 were used to quantify the model bias in simulations with and without forest parameterization (see following sections). At all towers only the respective highest available altitudes above ground level (AGL) were used, which means data at 30 m, 60 m and 100 m AGL on 30 m, 60 m and 100 m towers, respectively. In-situ observations are based on the quality controlled

5 minute tower data provided by the National Center for Atmospheric Research Earth Observing Laboratory (NCAR EOL) (UCAR/NCAR - Earth Observing Laboratory, 2019). Air temperatures were converted to potential temperatures by applying pressure observations at tower T47 (v01) in the valley at 2 m AGL as reference pressure $p_0$. Pressure at all used towers was then computed by means of $p_0$ and the hydrostatic equation for an isothermal atmosphere. This was necessary as pressure was not measured at each tower location.

---

[1]Experimental layout of the Perdigão campaign: https://perdigao.fe.up.pt/



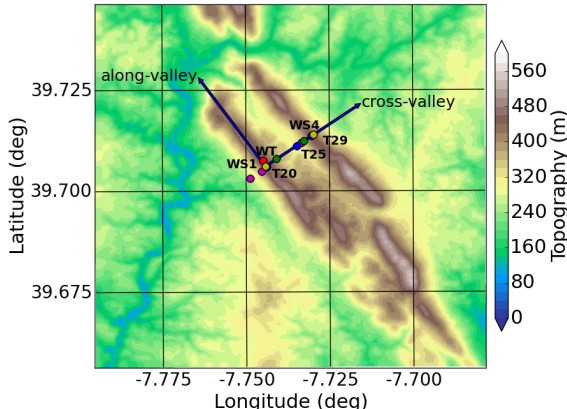

**Figure 1.** Topographic map of the Perdigão double ridge site. The shown area marks the modelling domain D4. The red dot indicates the

position of the wind turbine (WT) on the SW ridge. Blue, green and magenta dots mark the position of 100 m, 60 m and 30 m towers along

the southeast transect (TSE), respectively. Shown are the locations of the three 100 m towers T20, T25 and T29, the two 60 m towers T22

and T27 and the three 30 m towers T17, T18 and T26. Only 100 m towers are labeled. The DTU Doppler wind lidars WS1 and WS4 are

labeled with yellow dots, respectively. The lidars WS2 and WS3 are located close to WS1 and WS4 and are therefore not indicated. The blue

perpendicular arrows mark cross- and along-valley wind directions. Cross-valley winds are defined along the TSE (see text).

**Table 2.** WRF simulations used in this study: acronyms start with "ST" for short-term and with "LT" for long-term simulations and end with

"_NF" and "_F" if the forest parameterization is switched off or on, respectively.

| Simulation | No. of domains | Forested domains | LES domains |
|---|---|---|---|
| ST_NF | 4 | - | D3 & D4 |
| ST_F | 4 | D3 & D4 | D3 & D4 |
| LT_NF | 3 | - | D3 |
| LT_F | 3 | D3 | D3 |





**Table 3.** Short-term (ST) and long-term (LT) simulation cases.

| Simulation case | Simulation interval |
|---|---|
| ST 7 May LLJ NE | 18 UTC 6 May 2017 - 6 UTC 7 May 2017 |
| ST 8 May LLJ NE | 18 UTC 7 May 2017 - 6 UTC 8 May 2017 |
| ST 22 May LLJ SW | 18 UTC 21 May 2017 - 6 UTC 22 May 2017 |
| LT IOP | 0 UTC 30 April 2017 - 18 UTC 18 June 2017 |

## 2.3   WRF set-up

In this study, numerical simulations are performed with the Weather Research and Forecasting (WRF) model version 4.0.1 (Skamarock et al., 2008). It is distinguished between short-term (ST) and long-term (LT) simulations. The set-up of the LT simulations is the same as in Wagner et al. (2019). The set-up for the ST simulations is similar to the one used in Wagner et al. (2019), but uses four nested domains D1, D2, D3 and D4 with horizontal resolutions of 5 km, 1 km, 200 m and 40 m, respectively. D4 is plotted in Fig. 1, while D1 to D3 are shown in Wagner et al. (2019). Domain D1 and D2 are run in RANS (Reynolds Averaged Navier Stokes) mode, while domain D3 and D4 are run in LES mode. The high resolution of D4 is chosen to better resolve the double ridge. As in Wagner et al. (2019) vertical nesting is applied to define individual levels in the vertical for each model domain. For domain D1 to D4 36, 57, 70 and 82 vertically stretched levels are used and the respective lowest model levels are set at 80 m, 50 m, 15 m and 10 m above ground level (AGL). The model top is defined at 200 hPa (about 12 km height). In D1 and D2 the Mellor-Yamada-Janjic turbulent kinetic energy (TKE) scheme (Mellor and Yamada, 1982) is used in contrast to the LT simulations (Wagner et al., 2019), where the YSU-scheme is applied. The other physics parameterizations are the same as in Wagner et al. (2019). Initial and boundary conditions are supplied by ECMWF operational analyses on 137 model levels with a horizontal resolution of 9 km and a temporal resolution of 6 hours. As in Wagner et al. (2019), the Global 30 Arc-Second Elevation (GTOPO30) digital elevation model and the U.S. Geological Survey (USGS) landuse data set are used for D1 and D2. For D3 and D4, the Advanced Spaceborne Thermal Emission and Reflection Radiometer (ASTER) topography data set (Schmugge et al., 2003) with a horizontal resolution of 30 m and the Coordination of Information on the Environment (CORINE) land cover data provided in 2012 with a horizontal resolution of 100 m was used. For ST and LT simulations a 5 and 10 minute output interval was applied for the LES domains (D3 and D4), respectively. To improve the boundary layer flow, a forest parameterization was implemented in WRF, which will be described in more detail in the





next section. All ST and LT simulations were run with (ST_F; LT_F) and without (ST_NF; LT_NF) forest parameterization. An overview of the different simulation types is given in Table 2. As Wagner et al. (2019) only described a LT simulation with forest (LT_F), a LT simulation without forest (LT_NF) was performed additionally in this study to quantify the effect of the forest parameterization over a 1.5 month period. ST simulations with four domains were run for 12 hours, while LT simulations were run for 49 days and 18 hours for the whole period of the IOP. In this study ST simulations were performed for three example LLJ cases for which lidar scans suggested a LLJ with wind maximum at approximately hub height of the WT. Such cases are particularly relevant for power and load estimations in site assessment of wind turbines. ST simulations were initialized at 18 UTC the day before the LLJ events. LT simulations were started at 0 UTC 30 April 2017. Table 3 summarizes the dates and time periods of the ST and LT simulations.

## 2.4   WRF forest parameterization

In LES roughness elements like patches of trees, forests and buildings are in the same order as the horizontal and vertical grid size and become more important compared to mesoscale simulations, where in most cases only the roughness length $z_0$ is used to characterize the rough surface. This is also the case for standard WRF-LES, where three-dimensional roughness elements like trees are not explicitly considered and just characterized by $z_0$, which is obtained from the landuse data set. The explicit treatment of forest friction in numerical models is, however, of special importance for the realistic development of wind profiles including inflection points over forested areas. In this study the forest parameterization according to Shaw and Schumann (1992) is applied in the WRF code to study its impact on boundary layer flows over forested and complex terrain. Following Shaw and Schumann (1992) the additional forest friction term $F_i$, which acts on the lowermost model levels is defined as

$$F_i = -c_d LAD |\boldsymbol{V}| u_i, \tag{1}$$

where $|\boldsymbol{V}|$ is the amount of the three dimensional wind vector, $u_i$ is one of the three wind components, $c_d = 0.15$ is a constant drag coefficient and $LAD$ is the leaf area density profile characterizing the trees. $LAD$ is dependent on the tree type and the





height of the trees. The tree type is defined by means of the leaf area index $LAI$. The $LAD$-profile is computed according to

Lalic and Mihailovic (2004) at all points where trees are present as

$$LAD(z) = L_m \left( \frac{h - z_m}{h - z} \right)^n exp\left[ n\left( 1 - \frac{h - z_m}{h - z} \right) \right], \qquad (2)$$

with

$$n = \begin{cases} 6 & 0 \leq z < z_m \\ 0.5 & z_m \leq z \leq h, \end{cases} \qquad (3)$$

where $h$ is the tree height, $L_m = (LAI/h)1.69$ is the maximum $LAD$ at height $z_m = 0.6h$ following Mohr et al. (2014),

who used a similar forest parameterization in WRF. Example $LAD$-profiles are plotted in Fig. 2 for various $LAI$ and a tree

height of 30 m. With increasing $LAI$ the $LAD$ of the tree top becomes more dominant. In WRF simulations the $LAI$ is

retrieved from the CORINE landuse data set. The $LAD$-profile is computed according to the above formula in regions that are

classified as forest. As the forest height is not known from the landuse data, a randomly uniformly distributed tree height of

30 m $\pm$ 5 m is used. This is slightly higher than the real tree height in Perdigão, which is about 15 to 20 m, but ensures that

at least the lowermost two to three model levels are located within the canopy layer. The $LAD$ is set to zero in regions that

are not classified as forest. Fig. 3a) shows the distribution of $LAI$ for forested regions in domain D3 as given in the CORINE

data set. The randomly distributed forest height is plotted in Fig. 3b). White regions mark areas without forest. In contrast to

observed forest distribution (Menke et al., 2019b), the double ridge is completely covered by trees in the model according to

the CORINE data set.

## 3   Results

In this study three LLJ cases are analysed by means of the short-term simulations ST_F and ST_NF to investigate the impact of

the forest parameterization on the LLJ structure over the double ridge. The long-term runs LT_F and LT_NF are then analysed

to study the impact of the forest on the boundary layer flow during the whole IOP of Perdigão. Wagner et al. (2019) showed

that LLJs occured frequently over the double ridge and were mostly thermally driven night-time events blowing from NE.

LLJs from SW were also observed and allowed to study the dispersion of the wind turbine wake within and over the valley

(Wildmann et al., 2018a, b). We chose the NE-LLJ cases in the nights of 7 and 8 of May 2017 and the SW-LLJ case in the night





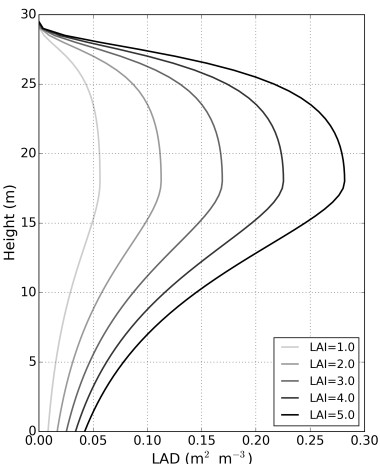

**Figure 2.** Leaf area density profiles $LAD$ for different example $LAI$ used to parameterize the forest drag. The $LAD$-profiles represent a pine tree canopy.

of 22 May 2017 to analyse the effect of the forest parameterization. These dates were selected as the jets were very stationary during these events. Table 3 gives an overview of the respective simulation periods. Fig. 4 shows lidar composites of the four DTU wind scanners WS1 to WS4 (see Fig. 1 for their locations). Plotted is the cross-valley, i.e. the horizontal wind component across the double ridge retrieved from radial velocities of the lidar scans for the three example LLJ cases. It can be seen that in all three LLJ cases gravity waves form over and in the lee of the double ridge. The two LLJs from NE indicate that the jet 5 structure can be very different. On 7 May the wind speeds are quite strong and the horizontal wavelength of the jet is the same as the wavelength of the double ridge (about 1.4 km). The agreement of the wavelength of the obstacle and the wavelength of the jet is consistent with a Froude number of 1.03, which was computed for this case according to Stull (1988) at 100 m AGL on the upstream mast T29. The wind direction was relatively constant with winds from NE near the surface over the double ridge and easterly winds above. On 8 May there are different layers of cross-valley winds induced by a strong wind shear with 10 height. The LLJ is weaker and thinner and shows shorter wavelengths, which are nearly half as long as the waves during 7 May. The Froude number for this case is 0.46, which agrees with shorter wavelengths of the jet. The wind shear is also visible in the wind barbs of the sounding indicating northeasterly winds within the jet and southerly and southeasterly winds above. The structure of the LLJ from SW in Fig. 4c) is similar to the NE jet in a), but was synoptically driven by a short-wave trough

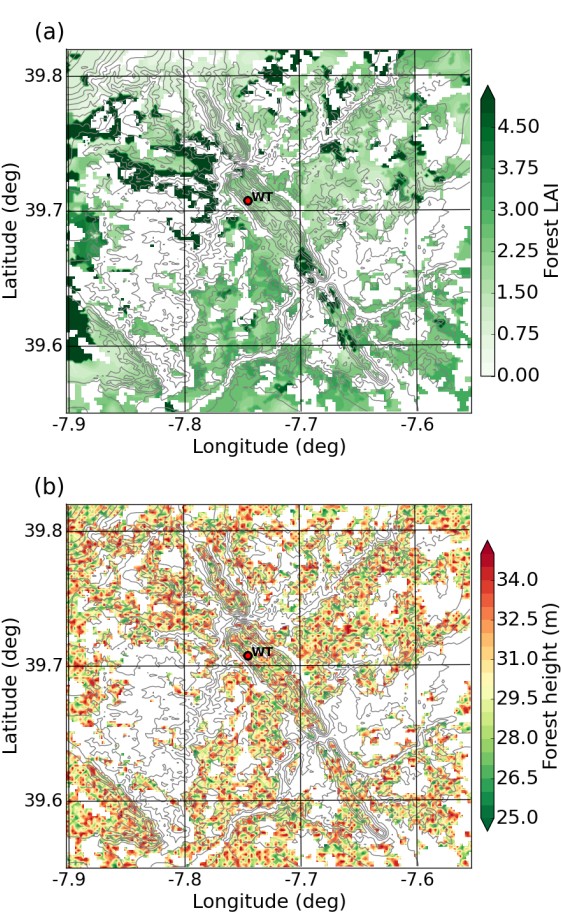

**Figure 3.** Distribution of (a) forest $LAI$ and (b) forest height for domain D3. White areas show regions that are not covered by trees. The

topography is indicated with grey contour lines (contour interval is 50 m) and the location of the wind turbine (WT) is marked with a red dot.



approaching the Portuguese coast from the West. With a Froude number of 1.06 computed at the upstream mast T20 longer wavelengths in the order of 1.4 km are confirmed. The sounding shows a wind shear from southwesterly winds over the double ridge to northerly winds at an altitude of about 1.5 km. All lidar cross sections indicate recirculation zones close to the surface both on the up- and downwind side of the double ridge and in the valley.

WRF ST runs were used to test the ability of the model to reproduce the flow situation over complex terrain. Fig. 5 to Fig. 6 and Fig. 7 to Fig. 8 show snapshots of cross-valley wind speed and potential temperature of ST_NF and ST_F simulations of both domain D3 and D4 for the same time as the respective measured cross-valley winds in Fig. 4. All simulations reproduce a LLJ, which approaches the double ridge from NE in a) and b) and from SW in c) and the simulated and observed soundings are principally in good agreement. The structure of the jets and especially the atmospheric layers directly over the surface are, however, very different, when comparing simulations with and without forest parameterization. Largest differences are visible when comparing ST_NF and ST_F simulations for D3 (Fig. 5 and Fig. 6) with the lidar observations in Fig. 4. Without forest parameterization (Fig. 5) wind speeds near the surface seem to be too high within the valley, over and on the lee side of the downwind ridge. The flow does not properly separate from the surface and recirculation zones are too weak or do not form at all. This is most obvious for the LLJ case on 8 May 2017 in Fig. 5b) where the ST_NF D3 run overestimates surface wind speeds downwind of the SW ridge. Lidar observations of the same case in Fig. 4 clearly show a flow separation from the surface including recirculation. It is known that all the above mentioned discrepancies can be partly traced back to wrong and reduced surface friction. It is for example shown in several studies that surface friction has a significant impact on the formation and wavelength of trapped lee waves (e.g. Richard et al., 1989; Jiang et al., 2006; Stiperski and Grubišić, 2011). It seems that for WRF-LES runs the representation of surface friction by the roughness length $z_0$ alone is not enough, as the grid size is in the order of obstacles on the earth surface like trees and buildings. The jet structure and the flow close to the surface agrees better with lidar observations when the forest parameterization is switched on in Fig. 6. Surface winds are reduced, recirculation zones develop and the amplitudes of the gravity waves agree better with lidar observations. The results are similar for the D4 simulations shown in Fig. 7 and Fig. 8 even if the ST_NF D4 runs without forest capture the lee wave structure of the LLJs considerably better than the ST_NF D3 run due to the better resolution of the topography. This means that qualitatively the differences between ST_NF D4 and ST_F D4 are reduced compared to the D3 runs. It becomes, however, clear that surface winds are reduced and the flow separation downwind of the double ridge is improved in runs with forest parameterization.

To quantify the effect of applying the forest parameterization, in-situ tower measurements of horizontal wind speed and potential temperature of all 30 m, 60 m and 100 m towers along transect TSE are compared to ST and LT simulations. WRF data

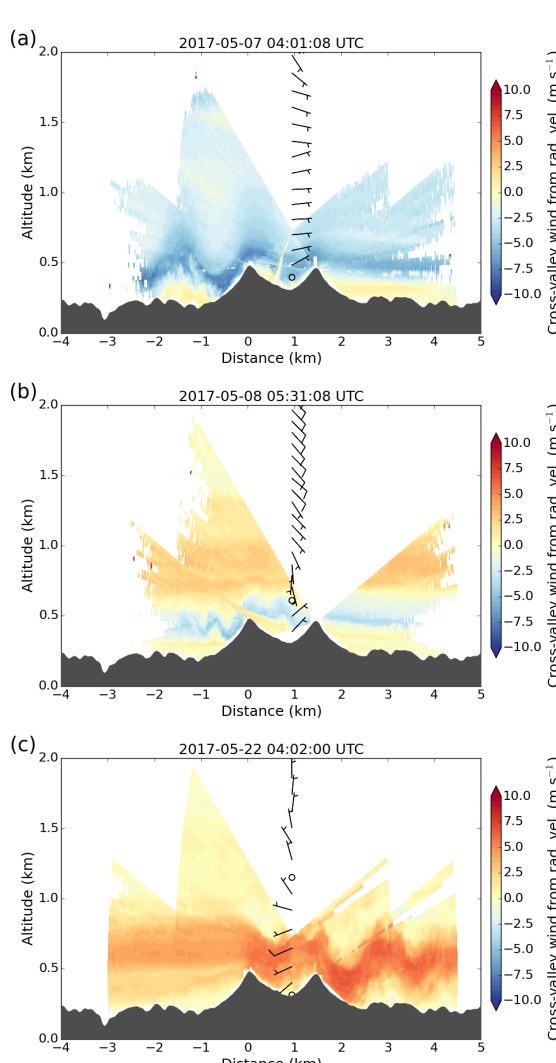

**Figure 4.** Cross sections of cross-valley wind speed (horizontal wind component across the double ridge) retrieved from radial velocity composite of DTU lidars WS1 to WS4 (colour contour interval: 0.5 m s$^{-1}$). Shown are LLJ cases from NE in a) on 07 May 2017 04:01:08 UTC and b) on 08 May 2017 05:31:08 UTC and a LLJ from SW in c) on 22 May 2017 04:01:09 UTC. The x-axis is centred at the location of WS1. Negative velocities indicate flow from NE. Black wind barbs indicate the horizontal wind speed and direction observed by radiosondes launched in the valley at 05 UTC every day. The vertical distance between the wind barbs is 100 m and the horizontal drift of the sounding balloon is not considered.



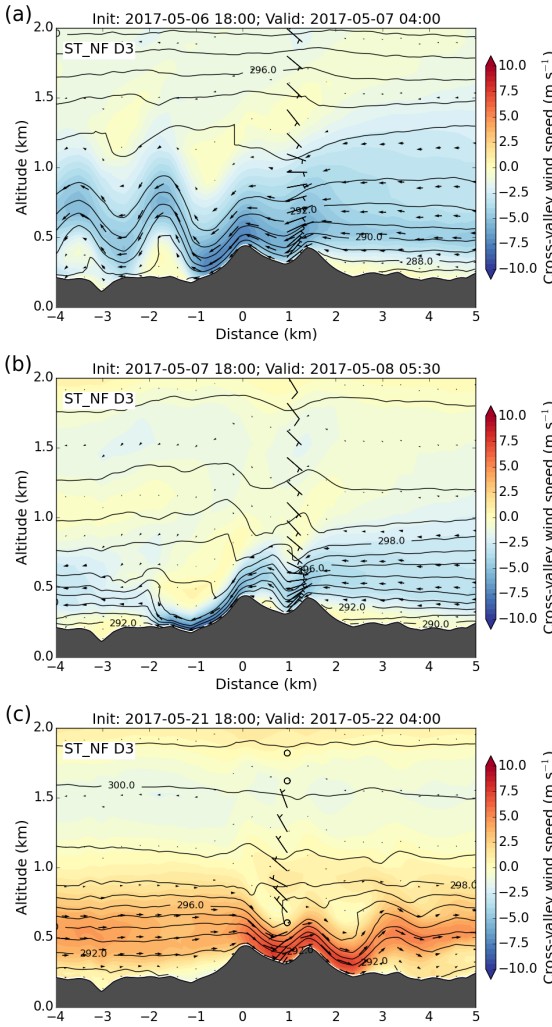

**Figure 5.** As in Fig. 4, but for the WRF ST_NF run without forest parameterization. Shown is the D3 cross-valley wind speed with colour contours (colour contour interval: 0.5 m s$^{-1}$), the 2-dimensional wind in the cross-section with arrows and the potential temperature with contour lines (contour line interval: 1 K). WRF data was interpolated to the same cross-section as in Fig. 4, which was defined by the location of WS1 and the azimuth scanning angle of 234.68°. The times of the respective LLJ-plots are virtually the same as in Fig. 4. The x-axis is centred at the location of WS1. Black wind barbs indicate the simulated horizontal wind speed and direction at the location of radiosondes launched in the valley at x=0.95 km and are plotted at every model level.



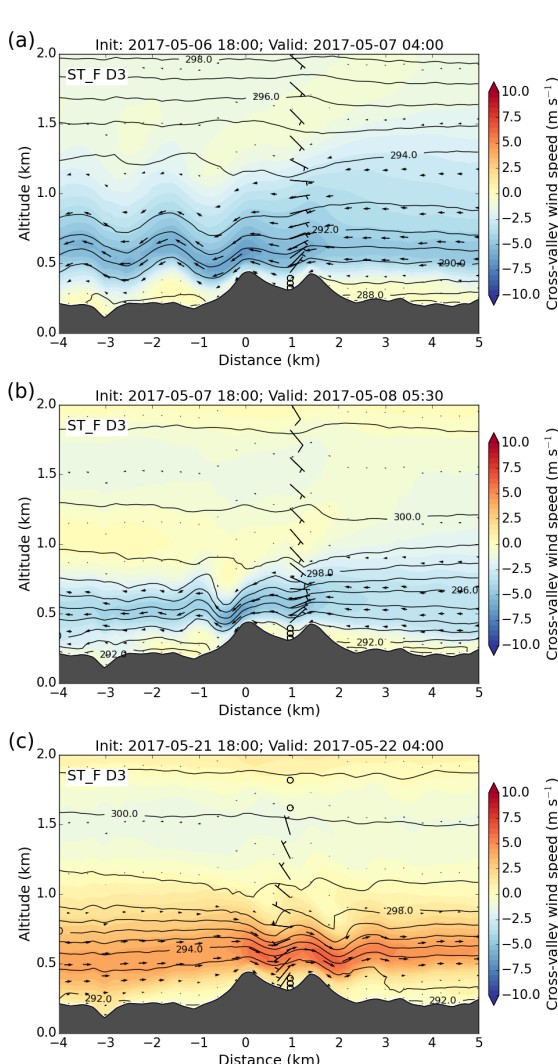

**Figure 6.** As in Fig. 5, but for ST_F D3 simulations (with forest parameterization).





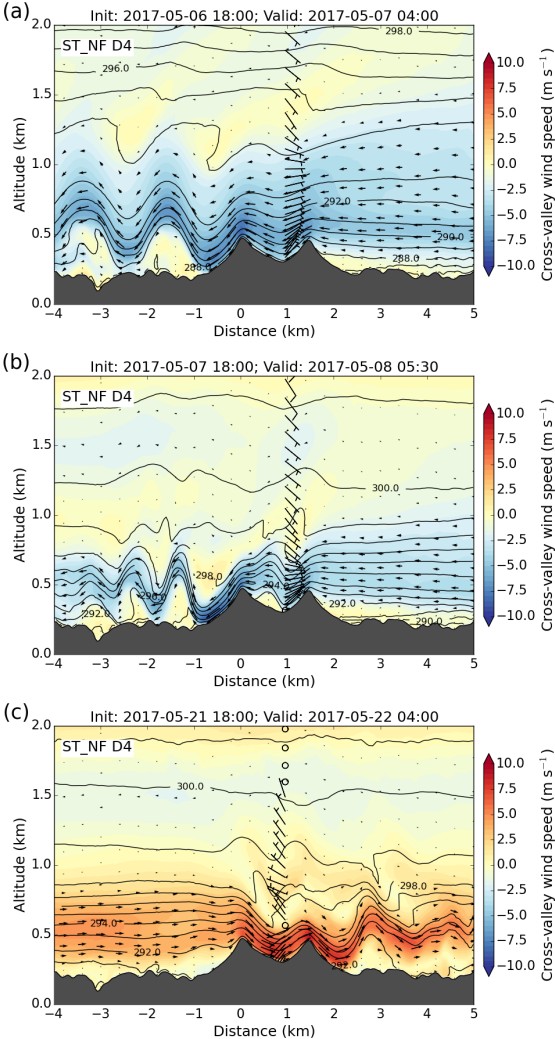

**Figure 7.** As in Fig. 5, but for ST_NF D4 simulations (without forest parameterization).



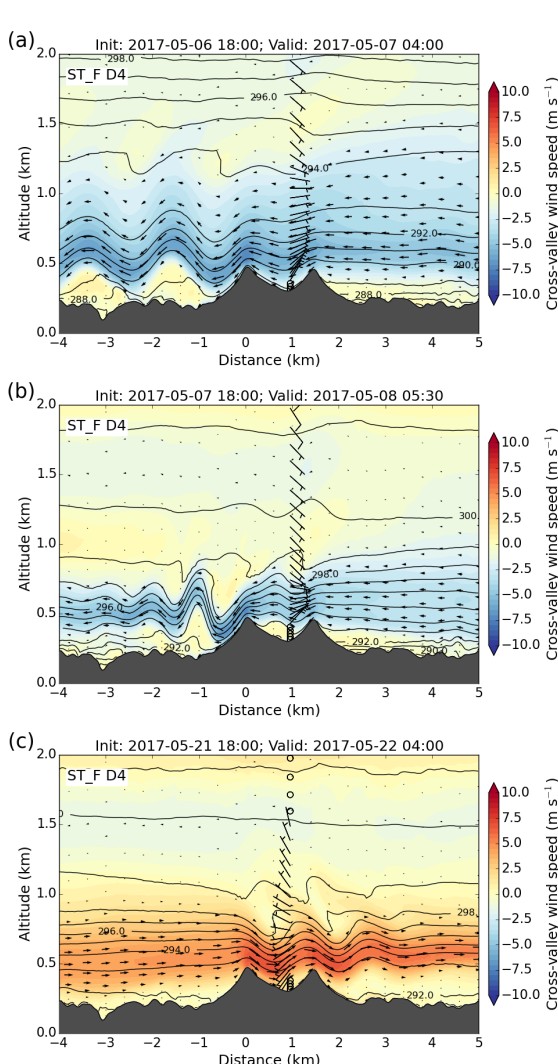

**Figure 8.** As in Fig. 5, but for ST_F D4 simulations (with forest parameterization).





is interpolated in space and time to the respective highest altitude of each tower. The comparison for ST simulations is started after a model spinup time of 3 hours, which results in a comparison interval of 9 hours (21 UTC to 06 UTC; see simulation intervals in Table 3). Tower data with a temporal resolution of 5 minutes are used for comparisons with ST simulations. 10-minute averaged tower data are used for comparison with LT simulations. Fig. 9 shows scatter plots of observed and simulated horizontal wind speeds at all towers for simulations with and without forest parameterization. The comparison of D3 5 simulations in a) indicates that ST_NF simulations overestimate near surface wind speeds. This was already seen qualitatively in the cross sections in Fig. 5. The implementation of the forest parameterization reduces the positive mean deviation (bias) of 1.19 m s$^{-1}$ to a bias of -0.85 m s$^{-1}$. The relatively bad Pearson correlation coefficient (cor) of 0.317 for the ST_NF D3 run is considerably improved to 0.648 for the ST_F D3 run and the root-mean-square error (rmse) is reduced by 0.74 m s$^{-1}$. For domain D4 the improvement of ST simulations with forests concerning correlation coefficient, rmse and bias is similar to 10 D3 runs as can be seen in Fig. 9b). Remarkable is the change of the bias from 1.02 m s$^{-1}$ for ST_NF D4 to -0.10 m s$^{-1}$ for ST_F D4.

As ST simulations were only done for LLJ events, the LT simulation of Wagner et al. (2019) was repeated without forest parameterization. This enables to quantify the effect of forest friction over a time period of 1.5 months. The comparison of surface horizontal wind speeds of the LT_NF and LT_F simulations with tower observations is plotted in Fig. 9c). As in the ST 15 simulations the distinct overestimation of wind speeds in the standard WRF set-up (LT_NF) and the considerable improvement in the LT_F simulation is visible. The extremely low correlation coefficient of 0.128 is changed to 0.715 when the forest is switched on. In addition, the rmse and bias is improved from 5.53 m s$^{-1}$ to 1.87 m s$^{-1}$ and from 3.97 m s$^{-1}$ to -1.03 m s$^{-1}$, respectively. The negative bias in both the ST_F D3 and the LT_F D3 simulations may indicate that the used tree heights of 30 m ± 5 m is too high and that improved landuse data with more realistic LAI and forest height distribution may be necessary. 20 An overview of all statistic values concerning horizontal wind speed are given in Table 4. The requirement of better landuse data becomes more clear when LT_NF and LT_F simulations are only compared to the two 100 m towers T20 and T29 on the ridges instead of including all towers along the slopes of TSE. Wind speed biases are reduced from 0.94 m s$^{-1}$ to -0.87 m s$^{-1}$ and from 1.28 m s$^{-1}$ to -1.24 m s$^{-1}$ at 100 m AGL and 80 m AGL (hub height of the wind turbine), respectively, when the forest parameterization is activated (see Table 4). This shows that there is a change from over- to underestimation of wind 25 speed when comparing the LT_NF and LT_F simulations and that the forest drag is too strong on the ridges in the LT_F run. Similar results are found in Menke et al. (2019b), where the same simulations are compared to lidar scans along the tops of both ridges. In this comparison wind speeds are too low in the LT_F simulation. The poor agreement of LT_F wind speeds





**Table 4.** Comparison of observed and simulated horizontal wind speed for short- (ST) and long-term (LT) simulations along transect southeast (TSE). Shown is the Pearson correlation coefficient, the root-mean-square error (rmse) and the mean deviation (bias). Values of simulations with forest are written in bold. The first three double-rows use all masts along TSE (see table 1). The last two double-rows only use data at T20 and T29 on the two ridge tops at 100 m AGL and 80 m AGL (hub height of the wind turbine), respectively.

| simulation | correlation | rmse (m s$^{-1}$) | bias (m s$^{-1}$) |
|---|---|---|---|
| ST_NF D3 @ all TSE masts | 0.317 | 2.61 | 1.19 |
| **ST_F D3** @ all TSE masts | **0.648** | **1.87** | **-0.85** |
| ST_NF D4 @ all TSE masts | 0.325 | 2.61 | 1.02 |
| **ST_F D4** @ all TSE masts | **0.592** | **1.84** | **-0.10** |
| LT_NF D3 @ all TSE masts | 0.128 | 5.53 | 3.97 |
| **LT_F D3** @ all TSE masts | **0.715** | **1.87** | **-1.03** |
| LT_NF D3 @ T20+T29 100m AGL | 0.486 | 2.88 | 0.94 |
| **LT_F D3** @ T20+T29 100m AGL | **0.610** | **2.32** | **-0.87** |
| LT_NF D3 @ T20+T29 80m AGL | 0.444 | 3.10 | 1.28 |
| **LT_F D3** @ T20+T29 80m AGL | **0.605** | **2.40** | **-1.24** |

with observations on the ridge tops can be explained by double ridges, which are covered by forest in the model (see Fig. 3), but are mostly free of trees in reality (see laser scan data in Menke et al., 2019b).

In Fig. 10, observed and simulated potential temperatures are compared. The effect of the forest parameterization is smaller than for wind speed and the correlation coefficients are in the order of 0.9 for both ST and LT simulations with and without forest friction. When looking at rmse and mean bias there is an improvement in the order of 0.06 K to 0.39 K and 0.2 K to 0.45 K, respectively, for both the ST and LT simulations with forest parameterization (see Table 5). When comparing LT_NF and LT_F simulations only to T20 and T29 on the ridges, rmse and mean bias values are in the same order as for comparisons with all towers along TSE (see Table 5).



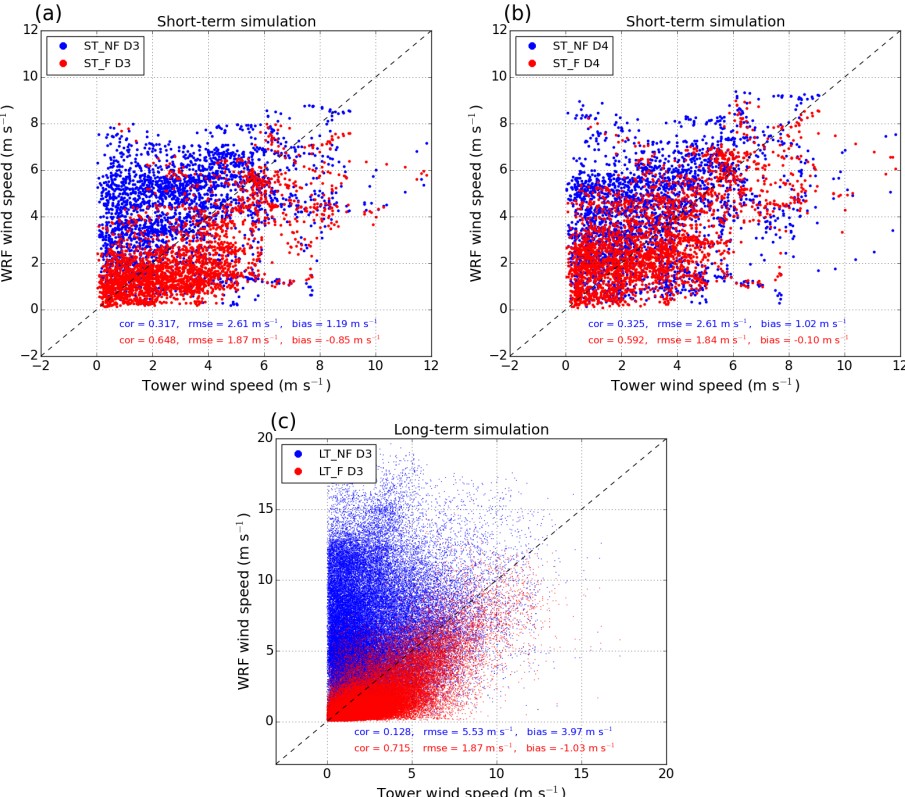

**Figure 9.** Comparison of simulated and observed wind speeds at all 30 m, 60 m and 100 m towers along the southeast transect (see Fig. 1)

for (a) and (b) short-term and (c) long-term simulations. Blue and red dots show data for simulations without (NF) and with (F) forest

parameterization, respectively (see also table 2 for simulation acronyms). For ST_NF and ST_F simulations all three LLJ cases are used and

shown for domain D3 in (a) and for domain D4 in (b). In (c) all data during the LT simulation period (30 April 2017 to 15 June 2017) are

used. The corresponding values for correlation coefficients (cor), root-mean-square-errors (rmse) and mean deviation (bias) are shown with

respective colours (blue: NF; red: F) in the plots. These numbers are also listed in Table 4.





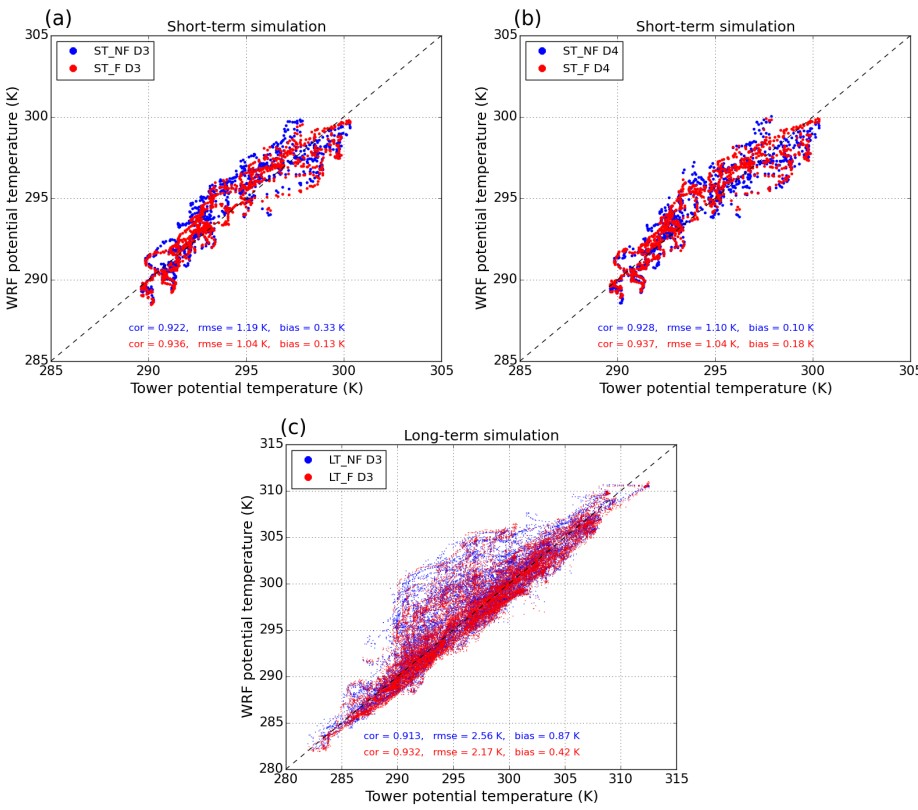

**Figure 10.** As in Fig. 9, but for potential temperature. Statistic numbers are also listed in Table 5.

## 4 Conclusions

In this study short- (ST) and long-term (LT) simulations were performed for characteristic LLJ cases and for the whole IOP of the Perdigão 2017 campaign. All simulations were run both with and without a forest parameterization in the LES domains and results were compared qualitatively and quantitatively to lidar cross sections and meteorological tower in-situ observations.

5 The comparison of ST simulations with lidar composites shows that standard WRF runs without forest parameterization do not reproduce the interaction of the LLJs with the complex double ridge topography and overestimate wind speeds near the surface. The missing forest friction in these runs prevents the formation of flow separation and recirculation zones in the valley as well as over and in the lee of the downwind ridge. With the forest parameterization, surface winds are reduced and recirculation zones and amplitudes of trapped lee waves agree better with lidar observations. The improvement is visible both in domain D3

10 and D4. The direct comparison of WRF data with in-situ observations of meteorological towers along the southeast transect





**Table 5.** As in Table 4, but for potential temperature.

| simulation | correlation | rmse (K) | bias (K) |
|---|---|---|---|
| ST_NF D3 @ all TSE masts | 0.922 | 1.19 | 0.33 |
| **ST_F D3** @ all TSE masts | **0.936** | **1.04** | **0.13** |
| ST_NF D4 @ all TSE masts | 0.928 | 1.10 | 0.10 |
| **ST_F D4** @ all TSE masts | **0.937** | **1.04** | **0.18** |
| LT_NF D3 @ all TSE masts | 0.913 | 2.56 | 0.87 |
| **LT_F D3** @ all TSE masts | **0.932** | **2.17** | **0.42** |
| LT_NF D3 @ T20+T29 100m AGL | 0.933 | 2.16 | 0.49 |
| **LT_F D3** @ T20+T29 100m AGL | **0.939** | **2.01** | **0.14** |
| LT_NF D3 @ T20+T29 80m AGL | 0.889 | 2.86 | 0.62 |
| **LT_F D3** @ T20+T29 80m AGL | **0.897** | **2.69** | **0.25** |

**Table 6.** Wind power density (WPD) computed at T20 on the SW ridge averaged over the whole IOP.

| data set | WPD @ 80 m AGL (W m$^{-2}$) | WPD @ 100 m AGL (W m$^{-2}$) |
|---|---|---|
| observation @ T20 | 127.41 | 136.03 |
| ST_NF D3 | 226.26 | 205.93 |
| **ST_F D3** | **67.41** | **96.89** |

(TSE) confirms the positive effect of the forest parameterization. The positive bias of horizontal wind speed is considerably reduced and the low correlations are improved (from 0.128 to 0.715 for the LT simulations). Simulated potential temperatures agree generally better with observations compared to wind speed and exhibit correlations in the order of 0.9 for both runs with and without forest parameterization. The forest parameterization has also a positive influence on potential temperature root-mean-square errors and mean deviations, which are reduced in all ST and LT simulations.

Wind speeds on the ridges can be transfered to wind power density $WPD = 0.5\rho u^3$ with air density $\rho$ and horizontal wind speed $u$, which is an important measure for site assessment of wind turbines. As there is a single wind turbine located on the SW ridge, wind power densities were computed by means of wind measurements at tower T20. The mean observed wind power





density at 80 m AGL and 100 m AGL at T20 is 127.41 W m$^{-2}$ and 136.03 W m$^{-2}$, respectively (see Table 6). Simulated wind power densities are overestimated by 77.58% and by 51.39% at 80 m AGL and 100 m AGL, respectively, in the LT_NF run. The LT_F simulation underestimates the power density by 47.09% and by 28.77% at 80 m AGL and 100 m AGL, respectively. This shows that the overestimation of wind power densities in the LT_NF run is stronger than the underestimation in the LF_F

run. Despite the negative bias of wind speeds in the LT_F simulation, correlation coefficients and rmse values are improved in simulations with forest parameterization when simulation results are compared only to data from T20 and T29 (see Table 4).

This analysis emphasizes that the representation of surface roughness elements like forests and buildings has to be improved in numerical weather models. The application of a roughness length alone is not enough to characterize the interaction of atmospheric flows with the surface and can lead to wrong flow structures and overestimated surface wind speeds. Similar

results were also found in Leukauf et al. (2019), who used a similar forest parameterization in WRF for simulations over a steep slope. In our study the additional forest friction was tested for LES runs, where horizontal and vertical grid sizes are in the order of roughness elements at the surface. For large forested areas it should be tested in further studies, if the forest parameterization has also a positive effect on RANS simulations with grid sizes in the order of 1 km. The LAD-profiles in this study were modelled with randomly distributed forest heights. The average forest height of 30 m was chosen for practical

reasons of the model setup. These heights exceed the actual forest height and the distribution does not represent the real coverage in the observed area. As a result, we find a negative wind speed bias in domain D3 and D4 caused by too large surface friction. This bias even increases when simulations are compared only to observations on the ridge tops. This was shown in this study by means of comparisons with tower T20 and T29 and is demonstrated in Menke et al. (2019b) by means of lidar ridge scans. More realistic forest distributions are only possible if landuse data sets are improved and provide up-to-date information

about forest coverage, forest height and LAI. Given the high sensitivity that we find for modelled surface wind speed and wind power density on the forest model, it is evident that a realistic surface representation in NWP models is essential for wind energy site assessment. Ideally, seasonal landuse data sets should be used in future, which account for changes in LAI, canopy height and coverage instead of using outdated static landuse data sets. Our findings show that at typical WT hub heights in the complex terrain site of Perdigão, the estimated wind power density error of a model output can be more than 75% depending

on the used surface drag parameterization.

**Code and data availability.** The WRF source code is available at: http://www2.mmm.ucar.edu/wrf/users/. The DTU lidar data are provided by Menke et al. (2018) and meteorological tower observations are supplied by UCAR/NCAR - Earth Observing Laboratory (2019).

**Author contributions.** Johannes Wagner performed the WRF simulations, implemented the forest parameterization and compared simulation results with observations. Norman Wildmann conducted lidar observations during the Perdigão campaign provided lidar data and helped to analyse observation data. Thomas Gerz took part in the Perdigão campaign and assisted in analysing the results. All authors contributed to the paper by discussing the results and by proofreading the manuscript.

**Competing interests.** All authors declare to have no competing interests.

**Acknowledgements.** We thank José Palma, University of Porto, José Carlos Matos and the INEGI team, as well as the research groups from DTU and NCAR for the successful collaboration and realization of the Perdigão campaign. Additionally, we thank the municipalities of Alvaiade and Vila Velha de Rodão in Portugal for local support. Thanks a lot to R. Menke and J. Mann from DTU for providing the wind scanner data and to NCAR EOL for the tower data. We appreciate constructive comments to the manuscript by two anonymous reviewers.

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
