# Peer review of "Improving boundary layer flow simulations over complex terrain by applying a forest parameterization in WRF"

_Wind Energy Science, 2019_

## Short Comment (SC1) · 13 Nov 2019

Dear authors,

This is a very interesting work. It will be very useful for the community that you share the WRF namelists you used for both the WPS and WRF runs (namelist.input and namelist.wps) at least for the case without forest parametrization to be able to reproduce your work. The description in your article of the simulations do not have all the details that one can find in the namelists. Also

Also, why not triggering turbulence? The flow patterns in such complex terrain might

change largely if turbulence is well developed, particularly with such a forest parame-
terization.

Regards,

---

## Author Comment (AC1) · 3 Dec 2019

Dear Alfredo,

thank you very much for your comment. We uploaded the used namelists for both the long and short runs as a supplement. You are right that triggering turbulence is important and should be included in future works. Especially, triggering turbulence at the boundaries of the LES domains could be done with a cell-perturbation method similar to the one described in Muñoz-Esparza et al. (2017). This method is, however, not included in the standard WRF repository. You are also right that triggering of subgrid-scale turbulence by the forest parameterization should be included in future

simulations. It goes, however, beyond the scope of this study to include the TKE-effect of the forest parameterization and to rerun all simulations.

Thanks and best regards,

J. Wagner and co-authors

References: Muñoz-Esparza, D., Lundquist, J. K., Sauer, J. A., Kosovic, B., and Linn, R. R.: Coupled mesoscale-LES modeling of a diurnal cycle during the CWEX-13 field campaign: From weather to boundary-layer eddies, J. Adv. Model. Earth Sy., 9, 1572–1594, https://doi.org/10.1002/2017MS000960, 2017.

Please also note the supplement to this comment:
https://www.wind-energ-sci-discuss.net/wes-2019-77/wes-2019-77-AC1-supplement.zip

---

## Referee Comment (RC1) · Anonymous Referee #1 · 18 Dec 2019

**Review of "Improving boundary layer flow simulations over complex terrain by applying a forest parameterization in WRF" by Johannes Wagner, Norman Wildmann, and Thomas Gerz, Manuscript number: wes-2019-77**

The manuscript describes the use of a canopy parameterization for characterizing surface roughness in WRF model simulations. Using the parameterization improves various aspects of the low-level flow in the challenging site of Perdigão, Portugal.

The topic of the manuscript is engaging and useful. The influence of forest roughness length on wind simulations has not been studied before in this way and with the WRF model.

The goals of the paper are exciting and worth publishing, but the methodology is generally not appropriate and, in many places, not well explained. Furthermore, the analysis of the model simulations lacks some depth. Two more significant points stand out:

Major issue 1). Performing 48-day simulations without re-initialization or nudging to control the error growth in the model domain is inappropriate. It does not conform to the best practices in atmospheric modelling. As recommended by Warner (2011), one should "understand the limitations to the predictability of the phenomena being modelled". After about 72 hours, the WRF simulations probably lost most of their predictability. Luckily, the prominent topographic features at Perdigão are a dominant control of the evolution of the flow. Thus, it is not essential if WRF captures all the details of the synoptic flow correctly. It is probably only necessary to get the right large-scale flow direction for the terrain to force the small-scale flow in the right way. However, the use of such long un-nudged simulations gives the wrong impression to the readers. The setup of the simulations is not appropriate for what the manuscript wants to show.

I understand that the goal of the paper is not to show that forest canopies improve weather forecasts (of course, it probably does). But calculating correlation coefficients that include all masts, different heights and different sites together give that impression. (I will come back to this point in comment 2). So, I suggest downplaying (or entirely removing) the correlations and focus on a more interesting analysis of the results.

Major issue 2). I think it a pity to combine all masts and heights in Figure 9. I believe there is much more to be understood. The added friction of the forest parameterization reduces the winds in places, but Figure 9 cannot show that. I think a more in-depth analysis of the results, via for example wind speed distributions at relevant sites and heights will be much more exciting and valuable. I don't see the point of the analysis of potential temperature either.

My recommendation is that the manuscript might be acceptable after significant revisions: explaining and supporting the decisions made in the model setup and expanding the analysis of the long simulations.

In addition to the significant issues above, I also have some minor comments:

1. Page 2, L4 and elsewhere: should be "real case", not "real-case". The dash is not necessary.

2. Page 2, L8: "This representation of surface friction becomes critical...". Which representation? It is not clear; please explain.

3. Page 2, L10: I would write "simulation", and not "forecasts".

4. Page 2, L20: The term "huge" is not a formal adjective for scientific writing. Please replace by "large".

5. Page 3, Table 1: TSE is not yet explained in the text.

6. Page 4, L9: "The four wind scanners WS1 to WS4 were performing range-height..." is the wrong verb tense. It should be "The four wind scanners WS1 to WS4 performed range-height..."

7. Page 4, L20-21: "National Center for Atmospheric Research Earth Observing Laboratory (NCAR EOL) (UCAR/NCAR - Earth Observing Laboratory, 2019)" is defined here, but already used in L5.

8. Page 5, Figure 1. Sorry, but I cannot see the difference between the red and magenta dots. Maybe use a different symbol?

9. Page 6, L10. You write that you set the lowest model level to 80m and 40 m in D1 and D2. What was the rationale for that choice? PBL schemes are known to be sensitive to this height (see Shin et al. 2011), mainly because a height of 80 m could often be above the surface layer. Also, a model top of 200 hPa is too low for the possible convective atmosphere during the late sprint and not recommended by the WRF model developers (see WRF Best Practices, link below).

10. Page 6, L11-12: Do I understand that the ST and LT simulations did not use the same PBL scheme? Why? Is the choice of PBL scheme in the inner LES domains important? Please justify with appropriate references the WRF parameterizations used.

11. Page 6, L15-19: You do not give enough detail in the description of how the topography and land use of the WRF model domains were generated. Also, the provided namelists (which use USGS classes and gtopo30) do not match the description in the manuscript (CORINE and ASTER). What kind of filtering (interpolation type and smooth options) was used? How accurately the D3 and D4 terrain match the real topography of the Perdigão site?

12. Page 7, L22. "amount of the ... wind vector", maybe the "module" will be a better word?

13. Page 8, L6: I think the equation is better written as Lm = 1.69 (LAI/h). Constants should come first.

14. Page 8, L9-10: You write "... LAI is retrieved from the CORINE land use dataset". This statement is incorrect. CORINE is only a land-use dataset, and there are no LAI values associated with it. Do you mean that you used the WRF landuse table? Does this include the seasonal variations? Besides, there are several versions of the CORINE dataset. Please provide a reference. Also, WRF has only tables for USGS and MODIS land categories. How did you use the CORINE data?

15. Page 8. Why did you use a random perturbation to the height of the forest? Laser scans of the area of Perdigão exist and were part of the field experiment data. As far as I know, the data is available from the Univ. Porto.

16. Page 8. I am missing a section that describes how the WRF model output was processed to compare with the lidar and mast measurements. There is a line in Figure 5, but I would like to see more. For example, what height was used in the vertical interpolation?

17. Page 8, L15-16: "...the double ridge is completely covered by trees in the model ...". Do you mean trees cover the two ridges and the valley?

18. Page 11, L5-6: Why not write "Fig. 5-6 and Fig. 7-8 show snapshots of..."

19. Page 12, the caption to Figure 4: The colours are not contours, so you should not write "colour contour interval: 0.5 m/s". You provide a colour scale for that. Also, the time of the last cross-section does not match the one in the figure caption.

20. Figures 4-8: What is the source of the shaded hill at the bottom of each cross-section?

21. Page 21-22: The discussion regarding wind power densities in the conclusion section seems irrelevant at this point. Yes, winds are stronger without the forest parameterization, and they will give much larger power densities, there is no need to emphasize this one more time. A couple of sentences would be plenty.

References:

Warner, T.T., 2011: Quality Assurance in Atmospheric Modeling. *Bull. Amer. Meteor. Soc.,* **92**, 1601–1610, https://doi.org/10.1175/BAMS-D-11-00054.1

WRF Best Practices:
https://www2.mmm.ucar.edu/wrf/users/tutorial/201907/chen_best_practices.pdf

Shin et al. 2011: Impacts of the Lowest Model Level Height on the Performance of Planetary Boundary Layer Parameterizations, https://doi.org/10.1175/MWR-D-11-00027.1

---

## Referee Comment (RC2) · Anonymous Referee #2 · 20 Jan 2020

Review of "Improving boundary layer flow simulations over complex terrain by applying a forest parameterization in WRF" by Wagner et al.

The manuscript by Wagner et al., 2019 provides interesting research on a very recent topic on coupled meso-microscale simulations over forest using the Weather Research and Forecasting model that was enhanced by a forest parametrisation. In general the paper ist quite well written and presents novel and interesting research. However, I have two major and a number of minor points to be accounted for before I can recommend the publication as scientific paper in WESC.

Major Points:

[Figure]

1) Nudging of the simulations When looking at the namelists that the authors provided as supplementary material it becomes clear that an analysis nudging is not applied in neither the short nor the long simulations. In case of the short simulations this might still be meaningful but for the 1.5 months simulation, a nudging should be applied to reduce the model error growth. This could in particular also improve the correlation that is investigated for the long term simulations.

2.) Wind power density discussion. Especially in situations with a local jet, the wind power of an assumed turbine of a certain size instead of the wind power density is much more meaningful. The rotor is integrating the power over the rotor area. Thus, I suggest to calculate the wind power using a rotor equivalent wind speed method here.

Minor Points:

Page 1 - Line 10 (Abstract): low-level jet events over the double-ridge -> better: across?

Page 2 - Line 2: The correct forecast of boundary layer flows -> I suggest rewriting to: An accurate forecast of boundary layer flows and surface winds is of special interest for wind power assessments

Page 2 Li- Line 12-13: Several modelling and measurement studies.... analysed the effect of a forest parametrisation -> How can a measurement study analyse the effect of a parametrisation? I guess you mean validation studies?

Page 2 - Line 24: Conlusions and an outlook is.... -> ARE finally given

Page 3 - Caption Table 1: For towers only data at the respective highest available altitude are used. -> This sentence sounds very odd. I suggest to consider rewriting to something like: The data from the highest available measurement device were used in case of tower data...

Page 3 - Line 3-4: .. in complex terrain and was part ... -> I suggest: conducted / organized in the framework of the NEWA proejct

Page 3 - Line 14: ... which are mostly night-time phenomena... -> You should add "at the site". There are other LLJ like coastal LLJ that are NOT nighttime jets.

Page 4 - Line 8-9: The data repository of the field campaign is provided.... -> I had to read this line several times to understand that one needs to go to the website and search for "Perdigao Field Experiment (2019)". I suggest to add a real reference (in case of LaTeX "misc") to the respective website: http://re3data.org/repository/r3d100013152

Page 4 - Line 17: with and without forest parametrisation -> Better: With and without the (or a) forest parametrization

Page 4 - Line 19: In-situ observation... -> This is all based on mast data in case of your study only isn't it?. Then you can write: Met mast observations

Figure 1 - Size: ->I think the labels inside the figure are large enough. However, in the final publication I would recommend to increase the figure by about 50%.

Page 6 - Line 10: set at 80m, 50m, 15m and 10m above ground level (AGL). -> With the WRF eta levels I assume that adding the word "approximately" here makes sense.

Page 6 - Line 11: In D1 and D3 the Mellor-Yamada-Janjic turbulent kinetic energy (TKE) scheme... -> Why not the MYNN scheme?

Page 7 - Line 7: for power and load estimations in site assessment of wind turbines -> I suggest rewriting: and load estimations in the assessment of wind turbine sites....

Page 7 - Line 12 ... compared to mesoscale simulations, where in most cases only the roughness length is used to characterize the rough surface. -> I disagree with this sentence because the land use characteristics are also used to define e.g. the albedo and other properties of the surface. Rewriting to "characterize the roughness of the surfce" could help or being more precise here. This is also true for the following sentence. Is it really just the roughness length or are other surface properties also prescribed in WRF-LES?

Page 8 - Line 1: The tree type is defined by means of the leaf area index LAI. -> One sentence here what LAI is and describes could be very helpful for the reader.

Page 8 - Line 8: In WRF simulations -> I suggest rewriting to: In the WRF model, the LAI is retrieved from....

Page 8 - Line 10: As the forest height is not known... -> I suggest to add "for this site" somewhere in the sentence

Page 9 - Line 1: These dates were selected as the jets were very stationary during these events. -> How was this selection made? Based on which data? Visual inspection of the scanning lidar data?

Figure 3 - market with a red dot -> The red dot is very difficult to see in b) - maybe use a grey dot?

Page 11 - Line 9: ... directly over the surface... -> ABOVE the surface?

Page 11 - Line 10: Largest diffeerences are visible -> I suggest adding a difference plot (panel plot) between the lidar and simulation data.

Page 11 - Line 17: It is for example shown... -> Doesn't this sentence repeat what the sentence before says and can be completely removed?

Figure 4: -> Is it really relevant to show the information above 1.5 km height here? Reducing this could give some more detail of the flow inside the LLJ.

Figure 5 - Caption: As in... -> This is really confusing as Figure 4 is measurement and 5 model data. I support using this to reduce repetitions but in this case it confuses from my point of view.

Page 17 - Line 3-4: Tower data with a temporal resolution of 5 minutes are used.... -> I suggest to use the same averaging intervals in both cases.

Page 17 - Line 5: ... at all towers for simulation... -> I think it would be helpful to add

the towers as vertical lines or dots in all panel plots.

Page 18 - Line 1: ... which are covered by forest in the model but are mostly free of trees in reality? -> Why? Couldn't you manually change the tree coverage in the model?

Figur 9 - Label in Figures: -> cor/rmse/bias are much too small, fontsize should be increased.

Page 22 - Line 13 - ... on RANS simulations with grid sizes in the order of 1 km. ->Should this really depend on the grid size or rather the distribution of layers in the lowest height above the ground?

Page 22 - Line 22ff - Ideally seasonal landuse data sets should bus used in the future... -> I support this idea. Can you name data sources for this? Do satellite data exist from which these seasonal land uses can be gerived.

Page 23 - Data Availability: -> I suggest to add information of the availability of boundary condition data (e.g. the ECMWF data) and share the namelists and the geo_em files via a github repository.

---

## Author Comment (AC2) · 23 Apr 2020

The comment was uploaded in the form of a supplement:
https://www.wind-energ-sci-discuss.net/wes-2019-77/wes-2019-77-AC2-supplement.pdf

---

## Author Comment (AC3) · 23 Apr 2020

**Improving boundary layer flow simulations over complex terrain by applying a forest parameterization in WRF**

**Reply to comments of anonymous referees of manuscript wes-2019-77**

Johannes Wagner et al.

April 7, 2020

**1  Introduction**

We thank both anonymous referees for their comments and acknowledge their effort to improve our manuscript. We are very sorry to inform you that in spite of the detailed suggestions of the reviewers we are not able to finish the manuscript. The reason for this is that the main author left the field of atmospheric research at the end of December 2019 and started a new job, which is not connected to research activity anymore. He was not able to finish the manuscript due to time reasons, which means that we will stop the review process and let the paper in its discussion mode. We are very sorry for this and want to thank the reviewers for their work and time. In the following we answered the major comments very shortly.

**2  Major comments reviewer 1**

1. Major issue 1). Performing 48-day simulations without re-initialization or nudging to control the error growth in the model domain is inappropriate. It does not conform to the best practices in atmospheric modelling. As recommended by Warner (2011), one should "understand the limitations to the predictability of the phenomena being modelled". After about 72 hours, the WRF simulations probably lost most of their predictability. Luckily, the prominent topographic features at Perdigão are a dominant control of the evolution of the flow. Thus, it is not essential if WRF captures all the details of the synoptic flow correctly. It is probably only necessary to get the right large-scale flow direction for the terrain to force the small-scale flow in the right way. However, the use of such long un-nudged simulations gives the wrong impression to the readers. The setup of

the simulations is not appropriate for what the manuscript wants to show. I understand that the goal of the paper is not to show that forest canopies improve weather forecasts (of course, it probably does). But calculating correlation coefficients that include all masts, different heights and different sites together give that impression. (I will come back to this point in comment 2). So, I suggest downplaying (or entirely removing) the correlations and focus on a more interesting analysis of the results.

$\Rightarrow$ **We agree that analysis nudging would generally improve the model results. To reduce boundary effects in our model simulations we chose a very large outer domain D1 (see Fig. 1a in Wagner et al. 2019) which allows the development of their own flow dynamics in the inner domains D2 and D3. At the boundarys of D1 ECMWF analysis data are used as boundary conditions every 6 hours to adapt the WRF solution to the ECMWF analysis. We also think that the effect of analysis nudging would not be so large, as the synoptic condition was dominated by calm high pressure systems during the IOP (see Wagner et al. 2019), which means that local flow systems could develop in the innermost domain D3, which are not affected much by the dynamics at the boundary of domain D1. In addition it was not possible to rerun all long-term simulations again with analysis nudging due to limited computational ressources.**

2. Major issue 2). I think it a pity to combine all masts and heights in Figure 9. I believe there is much more to be understood. The added friction of the forest parameterization reduces the winds in places, but Figure 9 cannot show that. I think a more in-depth analysis of the results, via for example wind speed distributions at relevant sites and heights will be much more exciting and valuable. I don't see the point of the analysis of potential temperature either. My recommendation is that the manuscript might be acceptable after significant revisions: explaining and supporting the decisions made in the model setup and expanding the analysis of the long simulations.

$\Rightarrow$ **Thank you very much for your comment. In our manuscript we wanted to show the effect of a forest parameterization on the flow over complex terrain and think that at the end the "total" effect of this parameterization has to be shown by comparing all masts. This shows the net-effect of the parameterization. We agree that it would also be interesting to distinguish between different masts to see why at some places there is a better or worse correlation. Due to limiations in personal ressources we, however, cannot do this. The idea concerning potential temperature was to see effects of different turbulent mixing at the surface and related differences in stratification due to the forest parameterization. We agree, however, that further analysis should have been done here.**

**3  Major comments reviewer 2**

1. Nudging of the simulations When looking at the namelists that the authors provided as supplementary material it becomes clear that an analysis nudging is not applied in neither the short nor the long simulations. In case of the short simulations this might still be meaningful but for the 1.5 months simulation, a nudging should be applied to reduce the model error growth. This could in particular also improve the correlation that is investigated for the long term simulations.

$\Rightarrow$ **We agree that analysis nudging would generally improve the model results. To reduce boundary effects in our model simulations we chose a very large outer domain D1 (see Fig. 1a in Wagner et al. 2019) which allows the development of their own flow dynamics in the inner domains D2 and D3. At the boundarys of D1 ECMWF analysis data are used as boundary conditions every 6 hours to adapt the WRF solution to the ECMWF analysis. We also think that the effect of analysis nudging would not be so large, as the synoptic condition was dominated by calm high pressure systems during the IOP (see Wagner et al. 2019), which means that local flow systems could develop in the innermost domain D3, which are not affected much by the dynamics at the boundary of domain D1. In addition it was not possible to rerun all long-term simulations again with analysis nudging due to limited computational ressources.**

2. Wind power density discussion. Especially in situations with a local jet, the wind power of an assumed turbine of a certain size instead of the wind power density is much more meaningful. The rotor is integrating the power over the rotor area. Thus, I suggest to calculate the wind power using a rotor equivalent wind speed method here.

$\Rightarrow$ **Thank you for this comment. We think that you are right and would have done these computations, if our personal ressources would have been available.**